# Autographa Californica Multiple Nucleopolyhedrovirus Enters Host Cells via Clathrin-Mediated Endocytosis and Direct Fusion with the Plasma Membrane

**DOI:** 10.3390/v10110632

**Published:** 2018-11-14

**Authors:** Fujun Qin, Congrui Xu, Chengfeng Lei, Jia Hu, Xiulian Sun

**Affiliations:** 1Wuhan Institute of Virology, Chinese Academy of Sciences, Wuhan 430071, China; qinfj@wh.iov.cn (F.Q.); xucr@wh.iov.cn (C.X.); cflei@wh.iov.cn (C.L.); hujia@wh.iov.cn (J.H.); 2University of Chinese Academy of Sciences, Beijing 100049, China

**Keywords:** AcMNPV, single-virus tracking, quantitative electron microscopy, clathrin-mediated endocytosis, direct fusion with the plasma membrane

## Abstract

The cell entry mechanism of *Autographa californica multiple nucleopolyhedrovirus* (AcMNPV) is not fully understood. Previous studies showed that AcMNPV entered host cells primarily through clathrin-mediated endocytosis, and could efficiently infect cells via fusion with the plasma membrane after a low-pH trigger. However, whether AcMNPV enters cells via these two pathways simultaneously, and the exact manner in which AcMNPV particles are internalized into cells remains unclear. In this study, using single-virus tracking, we observed that AcMNPV particles were first captured by pre-existing clathrin-coated pits (CCP), and were then delivered to early endosomes. Population-based analysis of single-virus tracking and quantitative electron microscopy demonstrated that the majority of particles were captured by CCPs and internalized via invagination. In contrast, a minority of virus particles were not delivered to CCPs, and were internalized through direct fusion with the plasma membrane without invagination. Quantitative electron microscopy also showed that, while inhibition of CCP assembly significantly impaired viral internalization, inhibition of endosomal acidification blocked virus particles out of vesicles. Collectively, these findings demonstrated that approximately 90% of AcMNPV particles entered cells through clathrin-mediated endocytosis and 10% entered via direct fusion with the plasma membrane. This study will lead toward a better understanding of AcMNPV infection.

## 1. Introduction

Baculoviruses, belonging to the baculoviridae, are enveloped DNA viruses that have been extensively applied for insect control and eukaryotic expression, and have potential applications in gene therapy [1,2,3,4,5,6]. Baculoviruses are a large family of rod-shaped enveloped DNA viruses that specifically infect insects. There are two diverse types of virion phenotype: budded virus (BV) and occlusion-derived virus (ODV). ODV initiates infection in the midgut cells of the host insects and is responsible for primary infection, while BV infects other tissues of the insect and is responsible for systemic or secondary infections [7]. *Autographa californica multiple nucleopolyhedrovirus* (AcMNPV), a model virus of the baculoviridae, has been extensively studied [8,9,10,11,12,13]. Further development of these applications requires further insight into the cell entry pathway. In this study, we investigated the cell entry pathway of the BV of AcMNPV.

The cell entry mechanism of AcMNPV is not fully understood [14]. Early immunoelectron microscopic investigations showed that AcMNPV enters insect cells via adsorptive endocytosis [15,16]. This was confirmed by research using inhibitor drugs, which showed that AcMNPV entered insect cells through a clathrin-mediated, low pH-dependent endocytic pathway [17]. However, other researchers revealed that AcMNPV could infect cells via direct fusion with the plasma membrane as evidenced by immunological microscopy observations [18]. This conclusion was further supported by the observation that AcMNPV could efficiently infect Sf9 cells in the presence of endocytosis inhibitors at low pH [19].

Enveloped viruses infect cells by binding to the receptors on the plasma membrane and manipulating the host cell for trafficking and replication. Clathrin-mediated endocytosis and fusion with the plasma membrane are the most common pathways that viruses adopt for infection [20,21]. During clathrin-mediated endocytosis, ligands bind to receptors at the cell surface and are then delivered to clathrin-coated pits (CCPs). The CCPs then recruit more clathrin to mature into clathrin-coated vesicles (CCVs) [22]. Later, the ligands and receptors are internalized into the cytoplasm, and delivered to early endosomes for sorting to different organelles [23]. By contrast, ligands that enter cells via direct fusion with the plasma membrane are independent of CCPs, CCVs, and early endosomes. Instead, the ligands are directly released into the cytosol.

Small GTPases of the Rab family are a central element of the trafficking machinery in the endocytic pathway, and regulate a series of vesicle trafficking events, including vesicle budding, transport, fission, and fusion [24]. In the cytoplasm, Rab GTPases are enriched in specific intracellular vesicles, which are of strategic importance in the determination of distinct endosome identity [25]. Of these, Rab5 and Rab7 are markers of early and late endosomes, respectively. During clathrin-mediated endocytosis, ligands are internalized via CCPs and CCVs, and are subsequently delivered to early endosomes [26].

Many viruses, such as SV40, influenza virus, vesicular stomatitis virus, and adeno-associated virus, infect cells via multiple pathways [22,27,28,29,30]. In certain cases, inhibiting one pathway only switches the virus to the alternative pathway, but does not have an obvious effect on overall infectivity, as determined using traditional biochemical techniques, making it challenging to study these multiple pathways [27]. In addition, the pathway which only a minority of virus particles adopt would be difficult to be detected in the biochemical assays using cell entry inhibitors. This issue can be overcome by single-particle tracking in living cells and quantitative electron microscopy [31,32,33]. It has been reported that AcMNPV enters insect cells primarily through clathrin-mediated endocytosis at normal pH, and can efficiently infect cells via fusion with the plasma membrane at low pH [19]. However, whether AcMNPV enters cells via these two pathways simultaneously at normal pH, and the exact manner in which AcMNPV is internalized into cells remains unclear.

In this study, by tracking individual AcMNPV in living cells expressing distinct fluorescent endocytosis-relevant markers, and quantitative electron microscopy of viral infection, we investigated the cell entry pathway of AcMNPV at the single particle level. The results showed that AcMNPV primarily entered cells through the clathrin-dependent pathway, with a small fraction entering through fusion at the cell membrane. AcMNPV virus particles were delivered to pre-existing CCPs, and inhibition of the assembly of CCPs with chlorpromazine (CPZ) significantly blocked the virus particles at the cell surface.

## 2. Materials and Methods

### 2.1. Cells and Viruses

Sf9 cells (Invitrogen, Grand Island, NY, USA) were cultured in Grace’s insect medium (Invitrogen) supplemented with 10% fetal bovine serum (Gibco, New York, NY, USA) [34]. The recombinant viruses (vAc^Pp10-EGFP^) were kindly provided by Dr. Xinwen Chen (Wuhan Institute of Virology, Chinese Academy of Sciences, Wuhan, China).

### 2.2. Construction of Clonal Cell Lines Expressing Endocytic Markers

The coding sequence of the full-length *LCa* gene (GenBank accession number: MK060106) was amplified from Sf9 cells. EGFP with a linker (GCTGCCGCCGCCGCTGCCGCCGCC) at its C-terminal was inserted into the pIB/V5-His vector (Invitrogen) using primers EGFP-F (CGCGGATCCAACTCCTAAAAAACCGCCACCATGGTGAGCAAGGGCGAGGAG) and EGFP-R (CGCGGATCCAACTCCTAAAAAACCGCCACCATGGTGAGCAAGGGCGAGGAG), resulting in pIB/V5^EGFP^ vector. The *LCa* amplified fragment was then inserted into the pIB/V5^EGFP^ vector using primers LCa-F (ATAGAATTCATGGATGATTTCGGAGACAATTTCG) and LCa-R (ATACTCGAGTCAAGCTACTTTAGTAGTGCGTGGTGG), resulting in pIB/V5^EGFP-LCa^.

Stable polyclonal cell lines were selected after the transfection of Sf9 cells with pIB/V5^EGFP-LCa^ in accordance with the manufacturer’s specifications (pIB/V5-His Vector Kit, Invitrogen). Briefly, the cell line expressing EGFP-LCa was constructed by transfection with plasmid pIB/V5^EGFP-LCa^, and was then selected with the antibiotic Blasticidin (Invitrogen). Millicell hanging cell culture inserts (Merck Millipore, Darmstadt, Germany) were used to isolate clonal stable cell lines as described previously [35]. The clonal cell lines were examined under confocal imaging, and the cell line that had the appropriate fluorescently-labeled marker was selected for western blotting analysis and live-cell imaging. The cell line Sf^EGFP-EGFP^ was analyzed using an anti-EGFP antibody (ProteinTech, Wuhan, China).

### 2.3. One-Step Growth Curve Assay

To test the influence of recombinant protein expression on BV production by the clonal cell line, vAc^Pp10-GFP^ was used to infect the clonal cell lines (Sf^EGFP-LCa^) at a multiplicity of infection (MOI) of 5. Culture medium was harvested at 12, 24, 36, 48, 60, 72, 96, and 120 h post-infection. All samples were stored at −80 °C before titers were determined by the end-point dilution assay in Sf9 cells. These experiments were performed in triplicate.

### 2.4. Live-Cell Imaging

Virus infection and purification were performed as previously described [36,37]. Purified viruses were resuspended in PBS containing 1% BSA and stored at 4 °C. Viruses were filtered through a 0.22-µm pore size filter (Merck Millipore, Darmstadt, Germany) immediately before confocal imaging. Cells were pretreated with CPZ (Sigma-Aldrich, St Louis, MO, USA) and ammonium chloride (Sigma-Aldrich). CPZ (20 µg/mL) and ammonium chloride (10 nM) were used to treat cells. Cells without drug treatment were set up as a control group. For drug treatment in live-cell imaging, cells were pretreated with Grace’s medium containing inhibitors at 27 °C for 30 min. Cells (Sf9^EGFP-LCa^ or Sf^EGFP-Rab5^) were infected with vAc^VP39-mCherry^ at a MOI of 50. The drug was maintained throughout the experiment. Cells were seeded on glass-bottomed culture dishes and grown to 50% confluence prior to infection for confocal imaging. Dishes were incubated in a heated chamber at 27 °C during live cell imaging. Single-virus tracking was performed as described previously [38]. Cells (Sf9^EGFP-LCa^), which were infected with vAc^VP39-mCherry^, were recorded at a rate of 90 timepoints per min. Cells (Sf^EGFP-Rab5^) infected with vAc^VP39-mCherry^ were recorded at a rate of 42 timepoints per min. The emission signal was collected using a 60× oil-immersion objective with a PerkinElmer UltraView VOX system. EGFP and mCherry were excited at 488 nm and 561 nm, respectively. For the emission of EGFP and mCherry, bandpass filters of 527/W55 nm and 615/W70 nm were used, respectively. The trajectory of fluorescent signals was analyzed with Imaris software (Bitplane, Zurich, Switzerland). The virus particles were tracked and analyzed with the spot detection and tracking function of Imaris (Bitplane, Zurich, Switzerland) [39]. Mean velocity and intensity of fluorescent signal were all calculated by Imaris.

### 2.5. Transmission Electron Microscopy

The virus stock was concentrated after ultracentrifugation and was then filtered through a 0.22-µm filter (Merck Millipore) immediately before infection to remove viral aggregates. The cells were treated with drugs as described above. Since viral particles could not be detected at the low doses, cells were infected at a high MOI as described before [40,41]. Sf9 cells were incubated with AcMNPV at a MOI of 1000 for 30 min at 4 °C, allowing the virus to attach to the cell surface. Cells were then incubated for 1 h at 27 °C. Cells were processed as described before [42,43]. Briefly, cells were fixed in 2.5% glutaraldehyde in PBS at 4 °C for 1 h. Ultrathin slices (80 nm) of these cells were examined by FEI Tecnai G^2^ 20 TWIN transmission electron microscopy.

## 3. Results

### 3.1. Construction and Characterization of Recombinant AcMNPV and Clonal Cell Lines

Studies have shown that EGFP-LCa fusion protein highlighted more than 97% of the CCPs and CCVs in live cells, while the fluorescent protein label did not compromise the functional integrity of clathrin molecules [22,44]. To investigate the cell entry of AcMNPV, a clonal cell line (Sf9^EGFP-LCa^) stably expressing EGFP fused to the light chain of clathrin (EGFP-LCa) was generated. To confirm expression of the fusion protein, the clonal cell line Sf9^EGFP-LCa^ was analyzed by Western blotting. An immunoreactive band of EGFP-LCa (51 kDa) was detected with the anti-EGFP antibody in the Sf^EGFP-LCa^ samples, but not in the negative control Sf9 sample (Figure 1A). One step growth curve of AcMNPV BVs in the clonal cell line Sf9^EGFP-LCa^ showed similar growth trends to the Sf9 cells, indicating that stable expression of the fusion protein had no significant effect on the production of progeny virus (Figure 1B).

In addition, another clonal cell line Sf^EGFP-Rab5^ and a recombinant nucleocapsid-labelled virus (vAc^VP39-mcherry^) were generated as previously described [45] and were also employed in these experiments. EGFP fused to the early endosome maker Rab5 was expressed in the Sf9 cells, resulting in clonal cell line Sf^EGFP-Rab5^. A second copy of the *vp39* gene fused with the *mCherry* gene was inserted under promoter p10 to generate the recombinant virus vAc^VP39-mcherry^ [45]. The appropriate incorporation of fusion protein into nucleocapsids preserved the infectivity of the virus while allowing individual virus particles to be tracked for an extended period of time.

### 3.2. AcMNPV Enters Cells via Clathrin-Mediated Endocytosis and a Clathrin-Independent Pathway

Many viruses infect cells via multiple pathways. Single-virus tracking in live cells provides mechanistic insights into the pathway of virus infection [31,46]. By tracking single-virus in living cell expressing fluorescent cellular markers, several viruses were demonstrated to enter cells via clathrin-mediated endocytosis and a clathrin-independent pathway [22,23].

To investigate how AcMNPV virus particles were internalized through CCPs, clonal cell line Sf^EGFP-LCa^ and recombinant virus vAc^VP39-mcherry^ were employed. Virus particles (vAc^VP39-mCherry^) were used to infect cells (Sf9^EGFP-LCa^), and fluorescent images were recorded by spinning disc confocal fluorescent microscopy. The EGFP-LCa signal showed discrete and dynamic fluorescent structures in live cells (Figure 2A), consistent with previous observations [22,44].

Cells (Sf9^EGFP-LCa^, green) were infected with virus (vAc^VP39-mCherry^, red), and selected frames from a virus particle delivered to a preformed CCP are shown in Figure 2A and Appendix A. Following attachment, the virus particle first moved along the cell surface before being delivered to a preformed CCP. After delivery to the CCP (indicated with an arrow), the fluorescence intensity of EGFP-LCa suddenly increased due to the beginning of its colocalization with CCP (Figure 2B). After being captured by CCP, the virus particle was trapped in the CCP, which made the mean velocity of the virus particle remarkably decreased (Figure 2C). Subsequently, the CCP would recruit more clathrin to induced maturation of the CCP toward a CCV, and the clathrin signal that colocalized with the virus particle continued to increase (Figure 2B), indicating that the virus particle induced maturation of the CCP toward a CCV. Thereafter, the fluorescence intensity of EGFP-LCa decreased rapidly to that of background, indicating the disassembly of the CCV (Figure 2B). These results were consistent with previous observations about dengue virus [47].

To investigate whether AcMNPV enters cells via multiple pathways, we performed quantitative analyses of AcMNPV internalization. The virus particles (vAc^VP39-mcherry^) were incubated with cells (Sf9^EGFP-LCa^) at 4 °C for 30 min, allowing the virus particles to bind to the cell surface but preventing internalization into the cytoplasm. CPZ, a reversible CCP inhibitor, prevents the formation of CCPs. Cells were incubated with virus particles in the presence or absence of CPZ, and then individual virus particles were tracked. Representative images of Sf9^EGFP-LCa^ cells infected with vAc^VP39-mCherry^ virus particles for 10 min are shown in Figure 2D. In the untreated cells, most of the virus particles were captured by CCPs, and internalized into the cytoplasm. By contrast, after cells were treated with CPZ, the majority of virus particles were blocked at the cell surface. These results showed that inhibition of CCP assembly significantly impaired viral internalization.

The results showed that inhibition of the assembly of CCPs with CPZ significantly inhibited the internalization of AcMNPV virus particles compared with control cells, indicating that CCPs are required for viral internalization (Figure 2E). The results also showed that, while about 81% of virus particles were observed to be internalized into the cytoplasm, only 72% of virus particles were found to be delivered to a preformed CCP (Figure 2F), indicating that about 9% of virus particles enter cells through a clathrin-independent pathway. These results implied that AcMNPV may enter cells via clathrin-mediated endocytosis and a clathrin-independent pathway.

### 3.3. Depolymerization of CPPs Inhibits the Delivery of AcMNPV Particles into Early Endosomes

After internalization, AcMNPV particles were delivered to early endosomes. Cells (Sf9^EGFP-Rab5^) expressing EGFP fused to the early endosome marker Rab5 were infected with virus vAc^VP39-mcherry^. Selected frames from a virus particle (vAc^VP39-mCherry^, red) delivered to early endosomes (EGFP-Rab5, green) are shown in Figure 3A and Appendix A. The virus particle underwent rapid movement until being delivered to an early endosome. After delivery to the early endosome, the fluorescence intensity of EGFP-Rab5 suddenly increased due to the beginning of colocalization with early endosomes (indicated with an arrow) (Figure 3B). When the virus particles were delivered to early endosomes, the virus particles were trapped in early endosomes, which made the mean velocity of the virus particle remarkably decreased (Figure 3C). Early endosomes undergoes homotypic fusion events to maturate into late endosomes [24]. Subsequently, the intensity of EGFP-Rab5 colocalized with the virus particle continued to increase (Figure 3C), indicating that the virus-bearing early endosome underwent fusion to mature. Thereafter, the fluorescence intensity of EGFP-Rab5 decreased gradually, indicating that the virus particle was gradually released from the early endosome (Figure 3C). Cells were incubated with vAc^VP39-mCherry^ in the presence or absence of CPZ.

Representative images of Sf^EGFP-Rab5^ cells infected with vAc^VP39-mCherry^ virus particles for 30 min are shown in Figure 3D. In the untreated cells, most of the virus particles were internalized into the cytoplasm and delivered to the early endosomes. By contrast, after cells were treated with CPZ, the majority of virus particles were blocked at the cell surface and were not transported to early endosomes. Quantitative analysis showed that inhibition of CCV assembly with CPZ significantly inhibited the internalization of AcMNPV virus particles compared with control cells (Figure 4A). During clathrin-coated endocytosis, virus particles were delivered to the early endosomes after internalization into the cytoplasm. As expected, treatment with CPZ also dramatically reduced the percentage of virus particles delivered to early endosomes (Figure 4B). These results implied that inhibition of CPP assembly impaired AcMNPV internalization and the delivery of virus particles to early endosomes.

### 3.4. AcMNPV Enters Cells via an Early Endosome-Independent Pathway

Virus particles that enter cells via clathrin-mediated endocytosis are delivered to early endosomes after internalization. In contrast, virus particles that enter via direct fusion with the plasma membrane are not delivered to early endosomes after internalization. Quantitative analysis showed that, in untreated cells, while about 82% of virus particles were observed to be internalized into the cytoplasm, only 74% were found to be delivered to early endosomes (Figure 4C), indicating that about 8% of virus particles enter cells through a clathrin-independent pathway. The results showed that, in CPZ-treated cells, about 48% of virus particles were observed to be internalized into the cytoplasm, while only 38% were found to be delivered to early endosomes (Figure 4D), indicating that about 10% of virus particles enter cells through a clathrin-independent pathway. Collectively, these results demonstrated that the majority of AcMNPV virus particles entered cells through clathrin-mediated endocytosis, with a small fraction entering via a clathrin-and early-endosome-independent pathway.

### 3.5. AcMNPV Enters Cells via Endocytosis and Direct Fusion with the Plasma Membrane

Tracking individual virus particles in living cells expressing EGFP-LCa and EGFP-Rab5 suggested that AcMNPV primarily entered cells through clathrin-mediated endocytosis, with a small proportion of viruses entering via a clathrin- and early endosome-independent pathway. However, the clathrin- and early endosome-independent pathway adopted by AcMNPV remained unclear. To address this issue, we investigated the cell entry pathway at the ultrastructural level. A series of successive viral infection stages were analyzed by transmission electron microscopy. Intact enveloped virus particles, delineated with a visible envelope, were observed by their electron-dense cigar-shaped structure in longitudinal sections or their round-shaped structure in cross sections. By contrast, unenveloped nucleocapsids, located in the cytosol and nucleus, were identified by their electron-dense structures that lacked a visible envelope. Vesicular stomatitis virus, which has a characteristic bullet-like shape and was similar to AcMNPV, had been investigated previously [40].

At first, AcMNPV particles bound to the cell surface (Figure 5A) and interacted with the cell surface (Figure 5B), indicating that spikes of virus particles had bound to receptors on the cell surface. Subsequently, the virus particles diverged to enter cells through two different pathways. AcMNPV particles were predominantly internalized by endocytosis. The virus particles were initially internalized with obvious invagination (Figure 5(C-i)). Later, the invagination was enlarged (Figure 5(D-i)) until the entire particle was completely invaginated (Figure 5(E-i)). Virus particles were then observed to be located in the extreme periphery of the cell (within a distance of 200 nm from the cell membrane), indicating that virus-bearing vesicles had been formed (Figure 5(F-i)).

In contrast to clathrin-mediated endocytosis, a minority of virus particles enter cells through direct fusion with the plasma membrane without obvious invagination. AcMNPV particles were fused with the plasma membrane with one end of the nucleocapsid in the cytosol (Figure 5(C-ii)). Later, release of the nucleocapsid into the cytosol was continued (Figure 5(D-ii)) until the nucleocapsid was almost completely released into the cytosol (Figure 5(E-ii)). Viral nucleocapsid was found to be located in the extreme periphery of the cell, indicating that nucleocapsid had been completely released into the cytosol (Figure 5(F-ii)). We also found that virus particles entered cells via two different pathways within one cell, i.e., a viral particle that was fused with the plasma membrane with one end of the nucleocapsid in the cytosol was observed (Figure 5(Gi)), along with virus-bearing vesicles that were located in the extreme periphery of the cell (Figure 5(Gii)). Subsequently, virus particles interacted with the inner surface of the vesicle (Figure 5H), indicating that fusion between the viral envelope and endosome membranes had been initiated. Then, the viral nucleocapsid was partially released into the cytosol, resulting in the formation of a hemifusion intermediate (Figure 5I)). Nucleocapsid was almost completely released into the cytosol with one apical head of the nucleocapsid still connected to the vesicles (Figure 5J). Finally, nucleocapsid was observed within the cytosol without any vesicles in the vicinity (Figure 5K).

### 3.6. Depolymerization of CPPs Inhibits AcMNPV Internalization and Inhibition of Endosomal Acidification Blocks Particles out of Vesicles

To investigate the role of clathrin and endosomal acidification in infection, we next performed quantitative ultrastructural analysis of AcMNPV entry using inhibitors. Ammonium chloride, a reversible endosomal acidification inhibitor, prevents the acidification of endosomes. Sf9 cells were pretreated with CPZ or ammonium chloride before infection. The inhibitors were present throughout the experiment. Sf9 cells without pretreatment with inhibitors were included as a control. Quantitative analysis of about 200 cell sections was analyzed for each group. All the virus particles observed, such as intact viral particles and nucleocapsids without envelope, were recorded for analysis. Three different stages of viral infection were observed and analyzed: internalization (attachment to the plasma membrane and invagination), trafficking in vesicles (single and multiple virus particles), and nucleocapsid in the cytosol.

The results showed that the majority of virus particles (89%) were internalized through invagination and CCPs, whereas a minority of virus particles (11%) were internalized through direct fusion with the plasma membrane without obvious invagination (Figure 6A). The results were consistent with population-based analysis of single-virus tracking. In addition, the results showed that inhibition of the assembly of CCPs with CPZ, significantly increased the proportion of virus particles for internalization, indicating that CPZ blocked AcMNPV infection at the initial stage of infection (Figure 6B). CPZ not only reduced the percentage of virus particles in vesicles (Figure 6C), but also decreased the proportion of nucleocapsids in the cytosol (Figure 6D). In contrast to CPZ, ammonium chloride did not obviously affect the percentage of virus particles for internalization (Figure 6B). Instead, ammonium chloride remarkably increased the proportion of virus particles in vesicles (Figure 6C), indicating that ammonium chloride blocked nucleocapsid release from vesicles. Since endocytic acidification was required for AcMNPV infection, inhibition of acidification with ammonium chloride was expected to inhibit nucleocapsid release. Correspondingly, ammonium chloride also significantly reduced the percentage of nucleocapsids in the cytosol (Figure 6D).

## 4. Discussion

Despite the extensive applications of AcMNPV, the exact manner in which AcMNPV virus particles are internalized into cells remains unclear. In this study, we investigated the host cell endocytic pathway of individual virus particles in live cells expressing distinct fluorescent endocytic markers. Our findings demonstrated that AcMNPV particles primarily enter cells through clathrin-dependent endocytosis, with a small fraction entering through direct fusion with the plasma membrane.

Early studies showed that AcMNPV fuses directly with the plasma membrane [18], while a later study using biochemical inhibitors suggested that AcMNPV enters insect cells predominantly through a low-pH-dependent endocytosis pathway [17]. Further studies revealed that AcMNPV could efficiently infect cells through direct fusion with the plasma membrane in the presence of endocytosis inhibitors after a low-pH trigger [19]. In our study, using the population-based analysis of single-virus tracking and electron microscopy, we revealed that most of the virus particles infected cells through clathrin-mediated endocytosis, with a small proportion of virus particles entering cells through direct fusion with the plasma membrane.

Viruses that enter cells through clathrin-mediated endocytosis are either delivered to preformed CCPs or induce new formation of CCPs. For example, most influenza virus particles first bound to the cell surface without any clathrin signal, then the clathrin signal was subsequently detected around the viruses and gradually increased [44]. Similar to influenza virus, infectious hematopoietic necrosis virus was also internalized through the de novo formation of CCPs [48]. This study revealed that AcMNPV particles first move along the cell surface, and are then delivered to preformed CCPs. The clathrin signal around AcMNPV particles increased after interacting with the CCPs, and the CCPs matured into CCVs. Then the clathrin signal colocalized with the virus particles gradually disappeared, indicating the disassembly of the CCVs. This model of association with clathrin is consistent with dengue virus, which has been demonstrated to be delivered to pre-formed CCPs [47].

Many viruses, such as SV40, influenza virus, and adeno-associated virus, infect cells via multiple pathways [22,27,28,29,30]. Single-virus tracking in live cells provides mechanistic and kinetic insights into the pathway of virus infection. For example, by tracking single fluorescently-labeled influenza virus particles in cells expressing EYFP-clathrin, influenza virus was demonstrated to enter cells via multiple pathways. While most of the virus particles predominantly infect cells through clathrin-dependent endocytosis, a small proportion of particles enter cells through a clathrin- and caveolin-independent pathway [22].

Viruses that enter cells through direct fusion with the plasma membrane, usually undergo a lipid mixing step, and then release genetic material into the cytosol. Fusion of the viral envelope with the plasma membrane or intracellular membranes prior to viral penetration is called lipid mixing. For human immunodeficiency virus type 1 (HIV-1), virus fusion with the plasma membrane does not progress beyond the lipid mixing step, and leads to abortive infection [20]. In this study, electron microscopy showed that a minority of AcMNPV virus particles not only underwent the lipid mixing step, but also released the nucleocapsid into the cytosol. In contrast to HIV-1, AcMNPV entered cells via direct fusion with the plasma membrane, and led to productive infection. In addition, while the same number of cell sections was observed in each group, the number of virus particles in cells treated with CPZ was decreased markedly by 46% compared with control cells, indicating that CPZ remarkably affected viral entry. Since CCPs were required for AcMNPV internalization in the initial stage of viral infection, virus particles that attached to the cell surface, but were not invaginated, may have been removed by washing when cells were processed for electron microscopy.

In summary, we propose the following model for cell entry of BVs of AcMNPV. AcMNPV predominantly enters cells through clathrin-dependent endocytosis, with a small fraction of viruses entering through direct fusion with the plasma membrane. At first, AcMNPV particles bound to the cell surface and were then predominantly delivered to pre-formed CCPs, and further invaginated to form virus-bearing CCVs. Inhibition of CCP assembly with CPZ significantly impaired viral internalization. After invagination into the cytoplasm, the CCVs were gradually disassembled and delivered to early endosomes. The virus-bearing endosomes underwent acidification, and the nucleocapsids were then released into the cytosol. When endosomal acidification was inhibited with ammonium chloride, the virus particles were blocked out of the endosomes. Importantly, a minority of virus particles were observed to fuse directly with the plasma membrane. After viral envelope fusion with the plasma membrane, the nucleocapsid was directly released into the cytosol (Figure 7).

These findings demonstrated that AcMNPV particles primarily entered cells through clathrin-mediated endocytosis, with a minority entering via direct fusion with the plasma membrane. Population-based analysis of single-virus tracking and quantitative electron microscopy provide mechanistic and kinetic insights into the route of cell entry. These findings will facilitate a better understanding of the infection pathway of AcMNPV and contribute to the rational design of recombinant protein expression, bioinsecticides, and gene therapy.

## Figures and Tables

**Figure 1 viruses-10-00632-f001:**
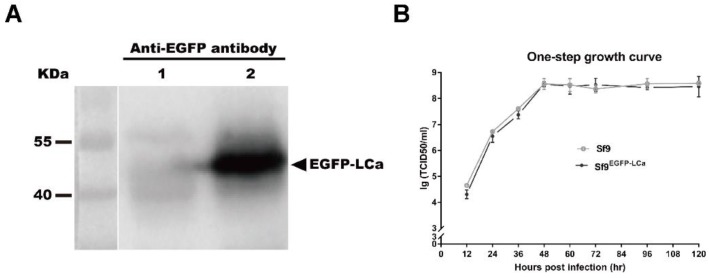
Construction and characterization of recombinant *Autographa californica multiple nucleopolyhedrovirus* (AcMNPV) and the clonal cell lines. (**A**) Western blotting analysis of cell lines Sf9 and Sf^EGFP-Rab5^ using an anti-EGFP antibody. An immunoreactive band of EGFP-LCa (51 kDa) was detected in the samples of Sf^EGFP-LCa^, but not in the sample of the negative control Sf9. (**B**) One-step growth curves of cell lines Sf9 and Sf^EGFP-Rab5^. Cells were infected with vAc^Pp10-EGFP^ at a MOI of 5. The titers of infectious budded virus were determined by TCID_50_ endpoint dilution assays.

**Figure 2 viruses-10-00632-f002:**
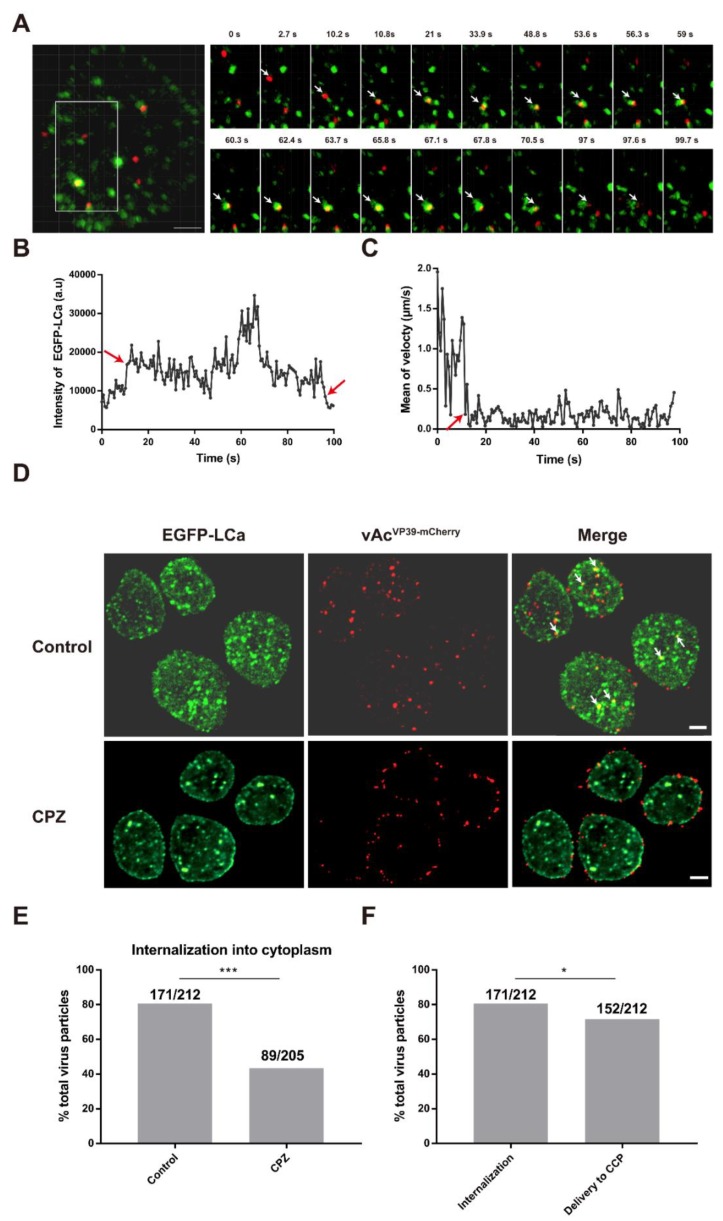
AcMNPV enters cells via clathrin-mediated endocytosis and a clathrin-independent pathway. (**A**) A representative example of a virus particle (vAc^VP39-mCherry^, red) delivered to a clathrin-coated pit (CCP) (Sf^EGFP-LCa^, green). When the intensity of EGFP-LCa decreased for more than five consecutive timepoints, the tracking of the virus particles will be terminated. Left panel: Fluorescent image of an EGFP-LCa-expressing cell (EGFP-LCa, green) infected with mCherry-labeled virus particles (vAc^VP39-mCherry^, red). The boxed region of the cell was magnified and shown on the right. Scale bar is 5 mm. Right panel: Selected frames from a virus particle (vAc^VP39-mCherry^, red) delivered to a preformed clathrin-coated vesicle (CCV) (EGFP-LCa, green). The virus particle is indicated with an arrow. The time is designated after viral attachment. (**B**) Time trajectories of the fluorescence intensity of EGFP-LCa colocalized with the virus particle. At 12 s post attachment (indicated with an arrow), the fluorescence intensity of EGFP-LCa suddenly increased, indicating that the virus particle was delivered to a pre-formed CCP. At 97 s (indicated with an arrow), the fluorescence intensity of EGFP-LCa decreased to that of background, indicating that the virus-bearing CCV was disassembled. (**C**) Time trajectories of the mean velocity of the virus particle. The virus particles first moved along the cell surface, and were then captured by a pre-existing CCP. After delivery to the CCP (indicated with an arrow), the mean velocity of the virus particle was remarkably reduced. (**D**) Representative images of EGFP-LCa vesicles (green) and virus particles (red) in live untreated and CPZ-treated cells. For the CPZ-treated group, cells (Sf^EGFP-LCa^) were pre-treated with CPZ for 0.5 h at 27 °C, and were then incubated with vAc^VP39-mCherry^ (MOI = 50) for 0.5 h at 4 °C. The temperature was raised to 27 °C, and the cells were imaged by time-lapse confocal microscopy. Images were captured as snapshots 10 min after infection. The virus particles that colocalized with EGFP-LCa-labelled CCPs are marked by arrows. Scale bar: 5 µm. (**E**) Quantification of the percentage of AcMNPV internalization into the cytoplasm in the absence or presence of CPZ. In the control cells, 171/212 represented the percentage of virus particles internalization into cytoplasm over total virus particles. In the CPZ-treated cells, 89/205 represented the percentage of virus particles internalization into cytoplasm over total virus particles. Virus particles that bound to the cell surface motionlessly or moved rapidly in a diffusive manner, were considered as invalid internalization. Individual virus particles were observed for 10 min upon binding to the cell surface (*** indicates *p* < 0.001, Pearson χ^2^ test). (**F**) Quantification of the percentage of AcMNPV internalization into the cytoplasm and delivery into EGFP-LCa positive CCPs. 171/212 represented the percentage of virus particle internalization over total virus particles. 152/212 represented the percentage of virus particle delivery to CCP over total virus particles. Virus particles that colocalized with the EGFP-LCa positive vesicles for more than 30 s were defined as indicating efficient delivery to CCPs. Individual virus particles were observed for 10 min upon binding to the cell surface (* indicates *p* < 0.05, Pearson χ^2^ test).

**Figure 3 viruses-10-00632-f003:**
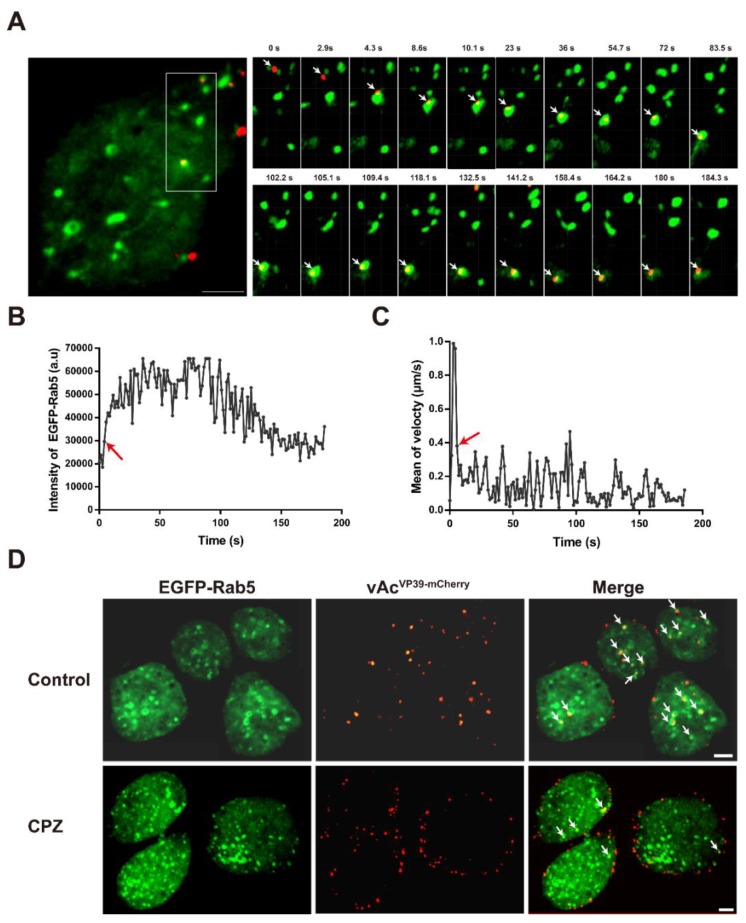
Depolymerization of CPPs inhibits the delivery of AcMNPV particles into early endosomes. (**A**) A representative example of a virus particle (vAc^VP39-mCherry^, red) delivered to an early endosome (Sf^EGFP-Rab5^, green). Left panel: Fluorescent image of an EGFP-Rab5-expressing cell infected with mCherry-labeled virus particles. The boxed region of the cell was magnified and shown on the right. Scale bar is 5 mm. Right panel: Selected frames from a virus particle (vAc^VP39-mCherry^, red) delivered to an early endosome (EGFP-Rab5, green). The virus particle is indicated with an arrow. (**B**) Time trajectories of the fluorescence intensity of EGFP-Rab5 colocalized with the virus particle. Immediately after being delivered to an early endosome, the fluorescence intensity of EGFP-Rab5 suddenly increased (indicated with an arrow). Subsequently, the intensity of EGFP-Rab5 colocalized with the virus particle continued to increase, indicating that the virus-bearing early endosome underwent fusion to mature. Thereafter, the fluorescence intensity of EGFP-Rab5 decreased gradually, indicating that the virus particle was gradually released from the early endosome. (**C**) Time trajectories of the mean velocity of the virus particle. The virus particle underwent rapid movement until being delivered to an early endosome. After delivery to the early endosome, the mean velocity of the virus-bearing endosome was remarkably reduced (indicated with an arrow). (**D**) Representative images of EGFP-Rab5 endosomes (green) and virus particles (red) in live untreated and CPZ-treated cells. For the CPZ-treated group, cells (Sf^EGFP-LCa^) were pre-treated with CPZ for 0.5 h at 27 °C, and were then incubated with vAc^VP39-mCherry^ (MOI = 50) for 0.5 h at 4 °C. The temperature was then raised to 27 °C. Images were captured as snapshots 20 min after infection. The virus particles that colocalized with early endosomes are marked by arrows. Scale bar: 5 µm.

**Figure 4 viruses-10-00632-f004:**
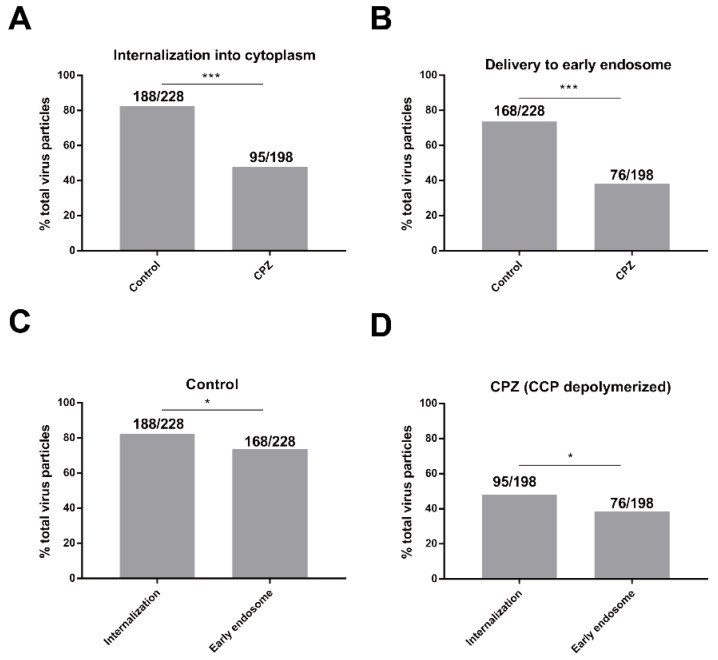
AcMNPV enters cells via an early endosome-independent pathway. (**A**) Quantification of the percentage of AcMNPV internalization into the cytoplasm in the absence or presence of CPZ. The percentage of virus particle internalization over total virus particles in control cells was represented by 188/228. The percentage of virus particle internalization over total virus particles in CPZ-treated cells was represented by 95/198. Individual virus particles were observed for 10 min upon binding to the cell surface (*** indicates *p* < 0.001, Pearson *χ*^2^ test). (**B**) Quantification of the percentage of AcMNPV delivery into early endosomes in the absence or presence of CPZ. The percentage of virus particle delivery to early endosome over total virus particles in control cells was represented by 168/228. The percentage of virus particle delivery to early endosome over total virus particles in CPZ-treated cells was represented by 76/198l. Virus particles that colocalized with the early endosomes for more than 1 min were defined as efficient delivery to early endosomes (*** indicates *p* < 0.001, Pearson *χ*^2^ test). (**C**,**D**) Quantification of the percentage of AcMNPV internalization into the cytoplasm and delivery to early endosomes in control cells (**C**) and CPZ-treated cells (**D**) The percentage of virus particle internalization over total virus particles in control cells was represented by 188/228. The percentage of virus particle delivery to early endosome over total virus particles in control cells was represented by 168/228. The percentage of virus particle internalization over total virus particles in CPZ-treated cells was represented by 95/198. The percentage of virus particle delivery to early endosome over total virus particles in CPZ-treated cells was represented by 76/198. (* indicates *p* < 0.05, Pearson *χ*^2^ test).

**Figure 5 viruses-10-00632-f005:**
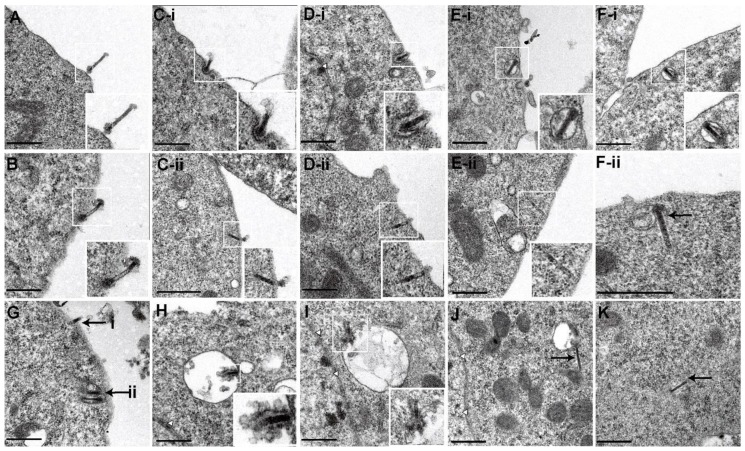
Ultrastructural analysis of the multiple pathway entry of AcMNPV into Sf9 cells using electron microscopy. AcMNPV particles were bound to Sf9 cells at 4 °C for 30 min, and then the temperature was raised to 27 °C for 1 h. (**A**) An enveloped virus particle that attached to the cell surface. The boxed regions of the virus particle are magnified and shown in the inset. (**B**) An enveloped virus particle that interacted with the cell surface, indicating that spikes of virus particles had bound to the receptors on the cell surfaces. The virus particles then entered cells mainly through two different pathways. Clathrin-mediated endocytosis was the primary pathway: (**C-i**) a virus particle that was partially invaginated into the cytoplasm; (**D-i**) a virus particle that was predominantly invaginated into the cytoplasm; (**E-i**) a virus particle that was almost completely invaginated into the cytoplasm; (**F-i**) a virus particle that was located in the extreme periphery of the cell, indicating that virus-bearing vesicles had been formed. In contrast to clathrin-mediated endocytosis, a minority of virus particles enter cells through fusion with the plasma membrane: (**C-ii**) a virus particle that was fused with the plasma membrane with one end of the nucleocapsid in the cytosol; (**D-ii**) nucleocapsid of a virus particle that was predominantly released into the cytosol; (**E-ii**) nucleocapsid of a virus particle that was almost completely released into the cytosol; (**F-ii**) a viral nucleocapsid that was located in the extreme periphery of the cell, indicating that the nucleocapsid had been completely released into the cytosol. (**G**) A viral particle that was fused with the plasma membrane with one end of the nucleocapsid in the cytosol (**i**), and virus-bearing vesicles that were located in the extreme periphery of the cell (**ii**). (**H**) An enveloped virus particle that interacted with the inner surface of a vesicle, indicating that fusion between the viral envelope and the endosome membrane had been initiated. (**I**) A viral nucleocapsid that was partially released into the cytosol, resulting in the formation of a hemifusion intermediate. (**J**) A viral nucleocapsid that was almost completely released into the cytosol with one apical head of the nucleocapsid still connected to the vesicles. (**K**) A nucleocapsid that was located in the cytosol without vesicles nearby. Scale bar, 0.5 µm.

**Figure 6 viruses-10-00632-f006:**
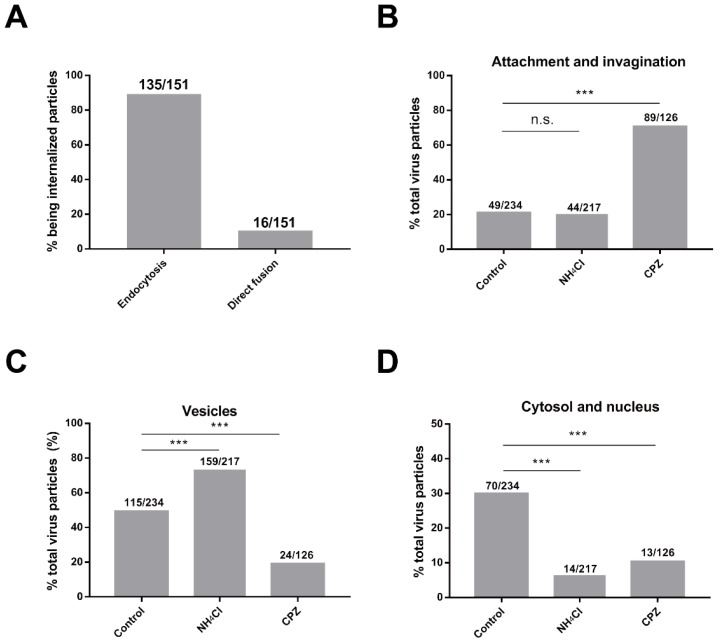
Quantitative ultrastructural analysis of AcMNPV entry into Sf9 cells. Sf9 cells were pretreated with CPZ or ammonium chloride (NH_4_Cl) before infection. The inhibitors were present throughout the experiment. Sf9 cells without pretreatment with inhibitors were included as a control. Quantitative analysis of about 200 cell sections was performed for each group. All the virus particles observed, such as intact viral particles and nucleocapsids without envelope, were recorded for analysis. (**A**) The percentage of virus particles that entered via endocytosis and direct fusion with the plasma membrane over the particles being internalized. (**B**–**D**) Three different stages of viral infection were observed and analyzed: the percentage of internalization (attachment to the plasma membrane and invagination, (**B**), trafficking in vesicles (single and multiple virus particles, (**C**), and the presence of nucleocapsid in the cytosol and nucleus (**D**). *** indicates *p* < 0.001, Pearson *χ*^2^ test.

**Figure 7 viruses-10-00632-f007:**
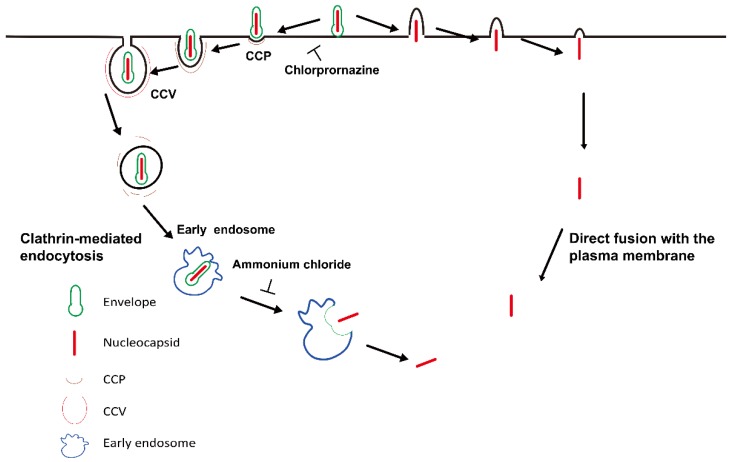
A proposed model for AcMNPV cell entry via clathrin-mediated endocytosis and direct fusion with the plasma membrane. AcMNPV particles first attached to the cell surface. They were then predominantly captured by pre-formed CCPs, and further invaginated to form virus-bearing CCVs. Inhibition of CCP assembly with CPZ significantly blocked viral internalization. After invagination into the cytoplasm, the CCVs were gradually disassembled and delivered to early endosomes. The virus-bearing endosomes underwent acidification, and the nucleocapsids were then released into the cytosol. When endosomal acidification was inhibited with ammonium chloride, the virus particles were blocked out of the endosomes. Importantly, a minority of virus particles fused directly with the plasma membrane. After viral envelope fusion with the plasma membrane, the nucleocapsid was directly released into the cytosol.

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
