# Peer review of "Autographa Californica Multiple Nucleopolyhedrovirus Enters Host Cells via Clathrin-Mediated Endocytosis and Direct Fusion with the Plasma Membrane"

_viruses, 2018, doi:10.3390/v10110632_

Round 1
Reviewer 1 Report
The authors show that a higher percentage of viruses enter cells by clatherin mediated endocytosis compared to direct membrane fusion using single-particle tracking and quantitative electron microscopy. Although the subject area is well covered in other literature the paper provides a granular analysis of virus entry into insect cells, pulling together existing hypotheses to draw a numerically based conclusion and model the percentage entry method. The paper is generally well written with a few grammatical errors.
Author Response
Dear reviewer,
We would like to thank you for the constructive comments and suggestions concerning our manuscript viruses-388117. Your comments were very insightful, allowing us to improve the quality of our manuscript. We have revised the grammatical errors in the revised manuscript.
We hope that the revisions to the manuscript are sufficient to make our manuscript suitable for publication in viruses. Thank you very much for all your help.
With my best regards,
Xiulian Sun, PhD
Wuhan Institute of Virology, Chinese Academy of Sciences

Reviewer 2 Report
This manuscript investigates the cell entry mechanisms of AcMNPV with the help of single-virus tracking and electron microscopy. The authors point out that it is known that AcMNPV enters cells via two pathways, clathrin-mediated endocytosis and direct fusion with the plasma membrane. They confirmed these known pathways. Additionally, the authors quantified the percentage at which viruses enter cells through these pathways by investigating these two pathways simultaneously. It is my understanding that the findings are new and that they add to the current understanding of AcMNPV cell entry. The experiments were conducted carefully with meaningful controls. Based on their findings the authors proposed a model for AcMNPV cell entry via two pathways that occur simultaneously.
I do have some questions and suggestions in methodology throughout the manuscript. Please find the attached file.

Author Response
Dear reviewer,
We would like to thank you for the constructive comments and suggestions concerning our manuscript viruses-388117. Your comments were very insightful, allowing us to improve the quality of our manuscript. We have revised the manuscript according to the comments.
We give our point-by-point response to your comments. We hope that the revisions to the manuscript and our accompanying responses are sufficient to make our manuscript suitable for publication in viruses. Thank you very much for all your help.
With my best regards,
Xiulian Sun, PhD
Wuhan Institute of Virology, Chinese Academy of Sciences
